# Unveiling Therapeutic Targets for Esophageal Cancer: A Comprehensive Review

Rakesh Acharya [1] , Ananya Mahapatra [1] , Henu Kumar Verma [2] and L. V. K. S. Bhaskar [1],*

1 Department of Zoology, Guru Ghasidas Vishwavidyalaya, Bilaspur 495009, India; rakeshacharya090@gmail.com (R.A.); biranchinarayanmahapatra0@gmail.com (A.M.)
2 Department of Immunopathology, Institute of lungs Health and Immunity, Comprehensive Pneumology Center, Helmholtz Zentrum, Neuherberg, 85764 Munich, Germany; henukumar.verma@helmholtz-munich.de
* Correspondence: lvksbhaskar@gmail.com or bhaskar.lvks@ggu.ac.in

**Abstract:** Esophageal cancer is a highly aggressive and deadly disease, ranking as the sixth leading cause of cancer-related deaths worldwide. Despite advances in treatment, the prognosis remains poor. A multidisciplinary approach is crucial for achieving complete remission, with treatment options varying based on disease stage. Surgical intervention and endoscopic treatment are used for localized cancer, while systemic treatments like chemoradiotherapy and targeted drug therapy play a crucial role. Molecular markers such as HER2 and EGFR can be targeted with drugs like trastuzumab and cetuximab, and immunotherapy drugs like pembrolizumab and nivolumab show promise by targeting immune checkpoint proteins. Epigenetic modifications offer new avenues for targeted therapy. Treatment selection depends on factors like stage, tumor location, and patient health, with post-operative and rehabilitation care being essential. Early diagnosis, appropriate treatment, and supportive care are key to improving outcomes. Continued research is needed to develop effective targeted drugs with minimal side effects. This review serves as a valuable resource for clinicians and researchers dedicated to enhancing esophageal cancer treatment outcomes.

**Keywords:** esophageal cancer; multidisciplinary approach; prognosis; molecular markers; targeted ttherapy

## 1. Introduction

Esophageal cancer (EC) is defined by an uncontrolled growth of tissues in the muscular hollow channel separating the throat from the stomach known as the esophageal wall. Esophageal squamous cell carcinoma (ESCC) and esophageal adenocarcinoma (EAC) are the two primary histological forms [1]. While EAC develops from glandular cells and mostly affects the lower esophagus, ESCC begins in the squamous epithelial cells lining the esophagus and primarily affects the upper and middle portions. Esophageal cancer is the seventh most prevalent cancer in terms of incidence and the sixth most common cause of cancer-related deaths globally [2]. According to recent studies, 90% of EC cases globally are caused by ESCC, which is particularly common in areas referred to as the "EC belt", encompassing sections of China, Iran, and central Asian republics [3]. On the other hand, EAC is more commonly observed in Western industrialized countries [4].

The risk factors for EAC include gastroesophageal reflux disease (GERD) and obesity, which is associated with the development or exacerbation of GERD [5]. Barrett's esophagus (BE), a condition in which the lining of the lower esophagus undergoes changes (metaplasia), is a strong risk factor for EAC [6]. In contrast, the main known risk factors for ESCC include tobacco smoking, alcohol consumption, consumption of pickled vegetables, hot food and beverages, and poor nutrition [7]. The primary treatment for EC is esophagectomy, or surgical removal of the esophagus. However, it is associated with significant mortality and morbidity rates, and many instances are detected at advanced stages, where surgery alone may not be sufficient [8]. According to recent studies, EGFR may be used as

a therapeutic target for EC [9]. Several medicines that hinder EGFR signaling have been developed, providing viable therapeutic options for EC. Biological medicines that target EGFR are being developed as new anticancer treatments [10]. Recently, most clinicians would recommend radical esophagectomy as the preferred treatment. Chemoradiation (CRT) is an alternative treatment, particularly for those who are unable to undergo surgery. In addition, the National Comprehensive Cancer Network recommendations propose ramucirumab, a VEGFR-2 antibody, in combination with paclitaxel, docetaxel, or irinotecan as a second-line treatment for unresectable or metastatic cancer. Interestingly, the KEYNOTE-590 trial will investigate the potential benefit of adding pembrolizumab to 5-FU plus cisplatin in patients with unresectable or metastatic EC [11]. In such circumstances, a multidisciplinary strategy combining surgical and non-surgical treatments is used to establish a cure. Endoscopic treatment is an option for mucosal cancer, but surgery is often suggested for resectable invasive tumors. Finding authorized medications that can be repositioned to treat EC patients is more time- and cost-effective than developing new ones for the treatment of different conditions [12]. Radiation therapy, chemotherapy, targeted medication therapy, and immunotherapy are also suggested depending on the stage and severity of the malignancy [8].

This review aims to provide readers with a comprehensive understanding of the therapeutic approaches employed in the management of EC. By examining these treatment options, we hope to contribute to the knowledge and understanding of this disease and its potential treatment strategies.

## 2. Local Treatment

### 2.1. Surgical Interventions

Squamous cell carcinoma (SCC) and adenocarcinoma (AC) are the two most common types of EC. Surgery has typically been the primary treatment for localized SCC and AC of the esophagus, particularly in those with early-stage illness. Surgery is essential in the treatment of resectable EC, particularly adenocarcinoma. Over the last three decades, advances in surgical techniques and radiation procedures have greatly improved clinical outcomes. Despite these major advances, survival rates with surgery alone for locally advanced EC remain low, with a median 5-year overall survival (OS) of only 20% [13]. For the treatment of EC, surgery is still the most effective single-modality therapy. The increased risk of relapse after esophagectomy has motivated researchers to look into multidisciplinary treatment options such as concurrent chemoradiotherapy (CCRT) with or without surgery [14]. However, the best treatment option for EC is still up for debate. Much research evaluating the curative potential of CCRT has cast doubt on the notion that surgery is an unavoidable aspect of curative therapy in the last decade [15].

It is feasible to have surgery to remove EC if it is discovered early and has not spread. The most common treatment for EC is surgery, although the extent of surgery varies greatly from patient to patient [16]. To diagnose remaining cancer cells after surgery, surgical treatment may be complemented with additional modalities such as chemotherapy and radiation therapy. To eradicate cancer cells, the procedure of surgery can be performed alone or along with other treatments.

The surgical approach involves the removal of a portion of the stomach, the segment of the esophagus affected by cancer, and approximately 3 to 4 inches (about 7.6 to 10 cm) of healthy esophagus above it if the cancer is located in the lower part of the esophagus, close to the stomach, or at the gastroesophageal (GE) junction where the esophagus and stomach meet. Afterward, the remaining esophagus is connected to the stomach either high in the chest or in the neck [17].

If the cancer is small, confined to the superficial layers of the esophagus, and has not spread, it can be removed along with the surrounding healthy tissue. This surgical procedure can be performed using an endoscope, which is inserted through the throat and into the esophagus.

### 2.1.1. Esophagectomy

For individuals without invasion of surrounding organs or distant metastasis, esophagectomy is considered one of the most effective treatments. Esophagectomy involves the partial or total removal of the esophagus. Surgery may be used to remove the esophagus and any nearby lymph nodes in order to treat the cancer if it has not spread significantly beyond the esophagus [18]. Unfortunately, in most cases, esophageal malignancies are not detected early enough for surgery to be successful. The three most commonly used techniques for thoracic EC are the transhiatal method, Ivor Lewis esophagectomy (right thoracotomy and laparotomy), and the McKeown procedure (right thoracotomy followed by laparotomy and neck incision with cervical anastomosis) [19]. It is worth mentioning that certain studies have shown that patients who undergo whole-piece esophagectomy have significantly higher survival rates compared to those who undergo transhiatal esophagectomy. This suggests that when the tumor is located in the lower esophagus or the cardia, whole-piece esophagectomy is preferred over transhiatal esophagectomy. The choice of surgical approaches and treatment strategies may vary based on the size and location of the tumor [20].

Esophagectomy can be performed using various methods. Regardless of the technique employed, esophagectomy is a complex procedure that often requires a prolonged hospital stay. It is crucial to have experienced professionals with extensive expertise in treating esophageal tumors and performing these operations [21].

### 2.1.2. Types of Esophagectomy
Open Esophagectomy

The traditional open technique for esophagectomy entails creating one or more significant incisions (cuts) in the neck, chest, or abdomen. The neck and abdomen are where the initial incisions are made during a transhiatal esophagectomy. A transthoracic esophagectomy, on the other hand, necessitates major incisions in the chest and abdomen [22]. In some treatments, the incisions are made in the neck portion as well as in the chest and in the abdomen.

Minimally Invasive Esophagectomy

Minimally invasive esophagectomy (MIE), which incorporates several surgical methods aimed at minimizing surgical stress, includes thoracoscopic/laparoscopic esophagectomy. Nowadays, MIE procedures include thoracoscopy/laparotomy, mediastinoscopy/laparoscopy, mediastinoscopy/laparotomy, and robot-assisted surgery with minimal incisions (RAMIE) [23,24]. With developments in endoscopic equipment and technology, surgically performed esophagectomy has grown in popularity [25]. For some early malignancies, multiple small incisions rather than one large one can be used to remove the esophagus. A laparoscope, a small, flexible tube with a light, is inserted through one of the incisions during surgery to provide visual guidance. Subsequently, surgical instruments are introduced through additional small incisions, allowing for precise removal of the esophagus [26]. To achieve successful outcomes with this treatment approach, it is crucial for the surgeon to possess exceptional skill and extensive experience in performing esophageal removal using minimally invasive techniques [27].

The findings indicate that both minimally invasive and open esophagectomy have similar effects on achieving radical tumor treatment [28]. Minimally invasive esophagectomy has demonstrated benefits in reducing intraoperative blood loss, postoperative hospital stays, pain, as well as the risk of pulmonary infection and vocal cord paralysis [29]. Therefore, it deserves clinical promotion and widespread application [30].

Lymph Node Removal/Dissection

Lymph nodes surrounding the esophagus are also removed during this form of esophagectomy. According to a study, a minimum of 15 lymph nodes are usually excised during surgical procedures [31]. The prognosis is less favorable if cancer has spread to

the lymph nodes, and the physician might advise additional treatments like chemotherapy or radiation therapy after the procedure.

Resection

Surgical resection has been the mainstay treatment for localized EC for several decades. Transhiatal esophagectomy and transthoracic techniques, such as Ivor Lewis esophagectomy (abdominal and right thoracic route, also known as Lewis–Tanner esophagectomy), are surgical options for resecting esophageal carcinoma [32].

Another method is the three-incision modified McKeown esophagectomy [33], which entails a laparotomy, right thoracotomy, and neck anastomosis, or a left thoracotomy or left thoracoabdominal approach [34]. The surgical procedure chosen is determined by the location of the tumor and the physician's discretion. All of these surgeries are complicated, and treatment at high-volume clinics with experienced surgeons and critical assistance with care has been linked to better results [35].

Risks and Complications of Esophagectomy

As with any surgery, there are risks associated with esophageal surgery. Anesthesia reactions, blood clots in the lungs or other areas, infections, and excessive bleeding are examples of short-term complications [36]. After surgery, the majority of people will experience some pain, which can typically be controlled with painkillers. Lung issues are quite common, and the possibility of developing pneumonia can result in a lengthy hospital stay and, in extreme cases, even death. Some individuals may experience changes in their voice following the surgery [37]. There is a chance of a leak forming at the point where the stomach (or intestine) connects to the esophagus, which could necessitate additional surgery to fix. However, due to advancements in surgical techniques, this occurrence is now less common. Some patients may experience swallowing issues due to strictures, which are narrowing conditions that can develop where the esophagus is surgically connected to the stomach [38]. These strictures can be addressed through an upper endoscopy procedure to alleviate the symptom. Surgery can sometimes damage the nerves responsible for stomach contraction, resulting in delayed gastric emptying, leading to regular nausea and vomiting [39]. Additionally, because the ring-shaped muscle (the lower esophageal sphincter) that normally keeps bile and stomach contents within the stomach may be removed or modified during surgery, there is a risk of bile and stomach contents flowing back into the esophagus, resulting in symptoms like heartburn. Antacids and motility medications can sometimes help manage these symptoms, but this surgery's complications can sometimes be fatal [40,41].

## 3. Treatments

Various treatments are available for managing esophageal obstruction and difficulty swallowing (dysphagia):

If the esophagus has been narrowed due to EC, the surgeon may opt to place a metal tube (stent) to keep the esophagus open. This procedure is performed using an endoscope and specialized tools. Other treatment options include surgery, radiation therapy, chemotherapy, laser therapy, and photodynamic therapy. In cases where swallowing difficulties persist after surgery, a feeding tube may be utilized. This tube allows for direct delivery of nutrition into the stomach or small intestine, enabling the esophagus to heal following cancer treatment [42].

### 3.1. Esophagogastrectomy

During an esophagogastrectomy, a significant portion of the stomach, as well as the esophagus and adjacent lymph nodes, is removed. The surgical procedure involves removing a portion of the stomach, the cancerous part of the esophagus, and roughly 3 to 4 inches (about 7.6 to 10 cm) of normal esophageal tissue above the tumor if the cancer is in the lower part of the esophagus close to the stomach or at the gastroesophageal (GE)

junction [43]. The rest part of the stomach is then raised and reattached to the esophagus. In some cases, a segment of the colon may be utilized for the connection [44].

Esophagectomy is a complex surgical treatment associated with significant rates of morbidity, mortality, and recurrence. The eligibility of a patient for surgical resection depends heavily on the extent of the disease and the patient's overall health. Accurate clinical staging plays a crucial role in determining the initial treatment plan for EC [45]. This includes a thorough clinical examination and computed tomography (CT) scans of the neck, chest, and abdomen for proper staging. Endoscopic ultrasound (EUS) and 18F fluorodeoxyglucose positron emission tomography-CT (PET-CT) should also be employed to assess the presence of lymphatic or distant metastatic disease in individuals with advanced malignancies who are potential candidates for surgical resection. Subsequently, a multidisciplinary team should determine the most appropriate treatment for each patient, taking into account factors such as tumor stage, location, histological subtype, comorbidities, and age [46].

*3.2. Endoscopic Treatment*

Endoscopic resection or endoscopic treatment can be used to treat early EC (EEC) that is limited to the mucosa layer and has not spread to lymph nodes (LNM) [47]. Early EC refers to tumors confined to the mucosa or submucosa without lymphatic dissemination or distant metastases. Advancements in endoscopic technology have increased the detection rate of early EC, with 31% of patients being diagnosed at an early stage [48]. Besides detection, endoscopic options have also been utilized for the treatment of an increasing number of early esophageal lesions.

Endoscopic resection (ER) is appropriate for lesions that are limited to the lamina propria or muscularis mucosae. On the other hand, due to the increased risk of lymph node metastases, patients with lymphovascular invasion of the submucosa or muscularis mucosae are not candidates for ER [49]. Endoscopic ablation (radiofrequency ablation, cryoablation, and photodynamic therapy), endoscopic mucosal resection, and endoscopic submucosal dissection are all options for endoscopic resection. Patients may not be eligible for ER if they have certain conditions, including large lesions (greater than 2 cm in size), Barrett's esophagus, and other esophageal conditions like varices [50]. Patients who are suited for ER need constant supervision for an extended period of time [51]. Esophagectomy may be recommended for patients who are unsuited for ER but are fit medically, while chemotherapy and radiation therapy may be considered for those who are unable to undergo surgery. Even though new research indicates that endoscopic therapy may be a secure substitute for patients with "high-risk" early EC (EEC), esophagectomy is still the advisable treatment for these individuals. In early EC, endoscopy therapy provides viable alternatives to traditional esophagectomy while causing significantly fewer complications. Endoscopic diagnosis and treatment represent the future trend in EC management, facilitated by advancements in endoscopic techniques.

When a patient is referred for endoscopic therapy of early EC, the tumor's stage and characteristics must first be determined. This is accomplished through a combination of endoscopic examinations and possibly other tumor progression modalities. Before making decisions regarding endoscopic therapy, a thorough examination of the lesion with the assistance of white-light endoscopy is essential. It is necessary to cleanse the esophagus to remove any liquids, food, or debris, followed by a comprehensive examination of the affected areas using white-light endoscopy. According to recent research, high-definition endoscopy surpasses standard-definition endoscopy in assessing mucosal changes in patients with Barrett's esophagus [52]. While white-light endoscopy remains the best method for assessing resectability, previous studies have explored additional approaches. One such approach is endoscopic ultrasound (EUS), which enables clinicians to determine the depth of the lesion and identify potential locoregional lymph nodes [53].

Curative therapy and palliative therapy are the two general categories of endoscopic management for EC [54]. Curative therapy is typically used for early ECs that are restricted

to mucosa and do not involve lymph nodes. When curative therapy is no longer feasible due to disease progression, the focus shifts to palliative care aimed at symptom improvement, particularly dysphagia. Endoscopic management in palliative care entails esophageal stent implantation, debulking, and dilatation [55].

### 3.2.1. Endoscopic Resection (ER)

Endoscopic Resection (ER) is the primary method for endoscopic management of early EC. For cases of high-grade dysplasia (HGD) and early EC, various Endoscopic Resection methods have been developed. Endoscopic Mucosal Resection (EMR) or Endoscopic Submucosal Dissection (ESD) are two methods for performing ER [56]. Both adenocarcinomas and squamous cell carcinomas (SCCs) can be treated with ER. According to the TNM staging of tumors (Table 1), the spectrum of conditions in which ER can be conducted in patients with adenocarcinoma often encompasses premalignant low-grade dysplasia in a patient with Barrett's esophagus (BE) up to, in rare situations, stage T1b adenocarcinoma. SCCs in patients with early EC staged as T1 or intramucosal can be treated with ER.

**Table 1.** TNM status and histologic grade definitions for EC depicted in 7th edition of American Joint Committee on Cancer (AJCC) Cancer Staging Manual.

| T status | |
| --- | --- |
| Tis | High-grade dysplasia |
| T1 | Invasion into the lamina propria, muscularis mucosae, or submucosa |
| T2 | Invasion into muscularis propria |
| T3 | Invasion into adventitia |
| T4a | Invades resectable adjacent structures (pleura, pericardium, and diaphragm) |
| T4b | Invades unresectable adjacent structures (aorta, vertebral body, tracheae) |
| N status | |
| N0 | No regional lymph node metastases |
| N1 | 1 to 2 positive regional lymph nodes |
| N2 | 3 to 6 positive regional lymph nodes |
| N3 | 7 or more positive regional lymph nodes |
| M status | |
| M0 | No distant metastases |
| M1 | Distant metastases |
| Histologic grade | |
| G1 | Well differentiated |
| G2 | Moderately differentiated |
| G3 | Poorly differentiated |
| G4 | Undifferentiated |

Currently used EMR methods typically allow for one-piece resection of lesions ranging from 15 to 20 mm. En bloc resection of larger mucosal lesions is possible with ESD. The choice between these methods is determined by the endoscopic equipment and the experience of the specialists. ESD is a promising alternative to EMR for the treatment of HGD and early-stage EC because it allows for endoscopic en bloc excision of lesions larger than 2 cm in diameter.

Several EMR techniques for endoscopic resection can be categorized on the basis of using a suction device or not. EMR is normally carried out using either the ligation- or cap-assisted technique. The cap-assisted technique involves affixing a specially created transparent plastic cap to the end of an endoscope, as first reported by Inoue and Endo [57].

The cap-assisted approach, commonly known as the "suck and cut" method, involves sucking the mucosa into a cap-fitted endoscope and then cutting the mucosa with a snare. Before suctioning, the snare is often opened as part of a pre-assembled kit. An alternative is to use a ligation device, which acts on the same "suck and cut" principles as the cap-assisted technique. This is the most often used approach in the United States. In the ligation-assisted technique, an instrument resembling a variceal band ligator is attached to the upper endoscope, and the mucosa is suctioned before a band is wrapped around it. Afterward, a snare is passed, and the band-supported mucosa is removed. Recent developments include the ability to advance a snare through the working channel of a standard endoscope, along with updated ligation cylinders equipped with multiple rubber bands, enabling endoscopists to perform multiple resections without the need to remove and reintroduce the endoscope [58]. Only small tumors smaller than 20 mm in diameter can be removed and blocked with tumor-free lateral margins, which appears to be the technique's principal drawback.

Evidence contrasting the two EMR approaches shows that they are, on the whole, comparable. The ligation-assisted method was faster with smaller resection specimens than the cap-assisted method, according to a randomized controlled experiment comparing the two procedures. However, the maximal thickness and adverse event rate of the resection specimens produced by both procedures were similar [59].

The ESD procedure involves dissecting the submucosal layer beneath the carcinoma using an electronic knife to obtain a large resection specimen with the neoplasm resected en bloc. This process involves three steps. First, the tumor is marked with electrocautery, then it is raised by injecting a saline solution beneath it, and finally, it is excised with an electrocautery knife. With the more current ESD approach, the targeted tissue is removed in one piece after carefully dissecting the submucosal lesions [60]. The drawback is that it takes longer and necessitates a deeper resection, which could enhance adverse effects even if it offers an en bloc specimen and can reveal information regarding resection margins [61]. While ESD had higher rates of curative resection and lower local recurrence rates than EMR, these advantages were outweighed by longer procedures and higher rates of bleeding and perforation according to a systematic review and meta-analysis comparing the two techniques [62]. Compared to EMR, some studies show that ESD may be linked to a higher risk of strictures and esophageal perforation [63].

In case of squamous cell carcinoma (SCC), ESD has shown superior outcomes in contrast to the EMR. According to the study's findings, if the diagnosis is SCC, EMR is typically thought to be suitable for minor lesions (10 mm or smaller); however, it is ideal for patients with bigger lesions to have ESD [64]. For the excision of Barrett's esophagus (BE) or early esophageal adenocarcinomas, current standards recommend EMR unless the lesions are larger than fifteen mm, do not lift well, or are at risk for submucosal invasion. In such cases, ESD should be performed. While EMR is suitable for smaller lesions, current guidelines typically advise ESD for individuals with squamous cell carcinoma (SCC) [65].

### 3.2.2. Advances in Endoscopic Therapy

Endoscopic therapy has become a viable option for individuals with neoplasia or early EC due to the low rates of lymphatic or hematogenous spread and the difficulties associated with esophagectomy. These treatments can be divided into two groups depending on whether they are used alone or in conjunction with other methods: ablative and non-ablative [66].

The use of ablative therapy is typically limited to flat lesions. Radiofrequency ablation (RFA) is the most popular type of ablative therapy. Cryoablation and photodynamic therapy (PDT) are other less popular techniques. Ablative therapy's main goal is to eliminate any precancerous or malignant tissue that may still be present in order to stop recurrence.

### 3.3. Radiofrequency Ablation (RFA)

Radiofrequency ablation (RFA) is the application of thermal energy directly to the esophageal mucosa. It generates thermal energy by using radiofrequency waves to destroy tissue. This can be performed with an endoscope-mounted device for more focused ablation or a circumferential ablation catheter. The delivered energy ensures uniform treatment to a depth of approximately 500 μm. As a consequence, the risk of stenosis is decreased because the treatment does not penetrate the submucosal layer [67].

RFA is typically advised for patients with intra-mucosal cancer, dysplasia, or non-nodular lesions, according to current recommendations. In patients who have undergone endoscopic resection (ER), it should be performed to treat any remaining Barrett's esophagus (BE). RFA's effectiveness in treating patients with BE or adenocarcinoma has grown, but its application to the treatment of squamous cell carcinoma (SCC) is still being explored. Recent research on early SCC has yielded encouraging results, with high rates of total eradication and low recurrence rates [68]. RFA is typically the most popular technique, and there is growing evidence to support its usefulness as well as a solid safety track record.

### 3.4. Photodynamic Therapy

PDT is an ablative procedure that causes mucosal destruction by activating a photo-sensitizer drug with laser light. PDT has been shown to be effective in the treatment of both esophageal adenocarcinomas and SCC [69]. PDT has received the most research attention of any ablative technique developed for the treatment of dysplasia and early EC. Intravenously or orally, a photosensitizing agent that is selectively absorbed by fast-growing cells, such as cancer cells, is administered. The photosensitizer is activated by applying an endoscopic laser directly to the malignant tumor [70]. But its serious side effects such as prolonged cutaneous photosensitivity, stricture formation, and recurrence have limited its use.

### 3.5. Cryotherapy/Cryoablation

Cryoablation has been considered for the treatment of pre-malignant and malignant esophageal problems [71]. Application of liquid nitrogen therapy is most frequently used. Using an open-tipped catheter, freezing carbon dioxide or liquid nitrogen is sprayed directly onto the tumor [72]. The targeted tissue is "frozen" and then thawed. This "freeze and thaw" cycle is repeated until the lesion is destroyed [73].

Endoscopic therapy is essential for treating both pre-malignant alterations and early-stage EC patients who have a low risk of lymph node metastases. Endoscopic technology's rapid advancement has improved current diagnostic and therapeutic capabilities for early EC. Endoscopic therapy has a lower morbidity and mortality rate than surgery, as well as equal rates of cure, rates of survival during five years, and a higher standard of living. In particular, post-operative complications from the treatment of EC can be managed with it. These fantastic outcomes are limited by the requirement for many treatments to achieve total eradication and the possibility of recurrences after eradication. Treatment success depends on patient compliance and careful patient selection using a multidisciplinary approach.

## 4. Systemic Treatment

### 4.1. Chemo-Radiotherapy for Esophageal Cancer

Chemotherapy (chemo) is a treatment that involves the administration of anti-cancer medications via injection or oral administration. The drugs travel through the bloodstream and reach cancer cells throughout the body. Chemotherapy rarely cures EC on its own, so it is frequently combined with radiation therapy, a treatment known as chemo-radiation [74]. Treatment options are determined by a variety of factors, including the type and stage of cancer, probable side effects, and the patient's preferences and overall health.

For tumors that have not spread beyond the esophagus and lymph nodes, combining multiple treatment methods is often prescribed. Chemotherapy, radiation therapy, and surgery are all options [75]. For squamous cell cancer, chemoradiotherapy is commonly

recommended as the initial treatment. Chemotherapy before surgery can benefit EC treatment in several ways. It can reduce the chances of cancer recurrence after tumor removal and shrink the tumor, making it easier to completely remove [76].

Patients with cT4b, significant lymph node metastases, or those who cannot undergo surgery are among the patients who should receive chemotherapy. When compared to surgery, it has been shown to be well-tolerated and effective. Chemoradiotherapy using docetaxel and cisplatin improved patients' median overall survival time to 23 months in a phase II trial [77]. For patients with unresectable EC (such as cancer in the upper third of the esophagus) and who are unable or ineligible for surgery, definitive chemoradiotherapy is the preferred treatment [78]. In terms of local tumor control, distant metastasis, and survival in patients with EC, concurrent chemoradiotherapy is superior to radiotherapy alone, but it is associated with more side effects [79].

In terms of definitive chemoradiotherapy, paclitaxel has been commonly used. Various combinations of radiation therapy and chemotherapy have been investigated. Neoadjuvant chemoradiotherapy (CRT) involves administering chemotherapy and radiation before surgery. Another treatment option is concurrent chemoradiotherapy (CCRT), where patients receive chemotherapy and radiotherapy simultaneously without undergoing surgery [80]. The Radiation Therapy Oncology Group (RTOG) conducted the RTOG 85-01 trial, which found that CCRT had the best results [81].

### 4.2. Radiation Technique

#### 4.2.1. Intensity-Modulated Radiotherapy (IMRT)

IMRT—In the neoadjuvant context, an appropriate radiation dose of 41.4–45 Gy is recommended [82]. Highly conformal treatment techniques, such as intensity-modulated radiotherapy (IMRT) or volumetric arc therapy (VMAT), have become the standard of care for treating EC. These techniques enable higher treatment doses while ensuring maximum organ protection, thereby reducing treatment-related side effects [83].

#### 4.2.2. Intensity-Modulated Radiotherapy

Three-dimensional conformal radiotherapy (3DCRT) is typically used to treat EC. However, compared to 3DCRT, IMRT with inverse planning provides a more accurate dose distribution to the target and sharper dose gradients around the target edges. With IMRT, a higher dose can be delivered to the target volume while sparing nearby healthy tissues [84]. Because EC is so close to the lungs and mediastinum, radiation can affect these organs, causing acute toxicities like radiation pneumonitis and late toxicities like cardiac events, pulmonary fibrosis, or radiation-related deaths.

#### 4.2.3. Volumetric Modulated Arc Radiotherapy (VMAT)

VMAT is an advanced technique that overcomes some of the disadvantages of IMRT. It reduces the number of monitor units (MU) and shortens the treatment time compared to IMRT [85]. VMAT can spare the lungs from intermediate-dose irradiation (V20, V30), but it may increase the volume of low-dose irradiation (V5, V10) [86].

#### 4.2.4. Proton Beam Therapy (PBT)

PBT is another radiation therapy option. PBT has the advantage of the Bragg peak, which allows for a significant reduction in radiation dosage to the surrounding normal tissues. Compared to IMRT, PBT reduces exposure to various volumes of the heart and lungs, particularly in cardiac parameters, indicating a lower risk of cardiac damage. This suggests that PBT could become one of the primary treatment methods for EC.

### 4.3. Targeted Drug Therapy for EC

EC is increasingly prevalent worldwide, with most cases being diagnosed at an advanced stage and having low survival rates despite current treatments. Chemotherapy has remained the primary treatment for EC for the past few decades without significant

advancements. Consequently, there is a critical need for new treatment options and approaches to improve outcomes. Targeted therapies have gained prominence due to the discovery of new biomarkers specific to EC [87]. These therapies focus on the genes, proteins, or tissue environment that contributes to cancer growth and survival [88]. By using drugs that target specific molecules, such as proteins found on or within cancer cells, targeted therapy disrupts the signaling pathways that regulate cell growth and division. This treatment approach effectively inhibits cancer cell growth and spread while minimizing harm to healthy cells. It is important to note that each tumor may have different targets, and targeted therapy is also referred to as molecular targeted therapy. For instance, the enzyme nicotinamide *N*-methyltransferase (NNMT), which is overexpressed in numerous neoplasms, can also be a promising therapeutic target for treating EC [89]. Macrocyclic peptides can be potent allosteric inhibitors of NNMT and can help in the management of EC [90].

In cases of locally advanced or metastatic adenocarcinoma tumors at the gastroesophageal junction, targeted therapy medications are often used in conjunction with chemotherapy.

These treatments may be considered as alternatives to surgery or for advanced esophageal malignancies that have not responded to other therapies. These targeted therapy medications, unlike chemotherapy, have specific mechanisms of action and primarily target cancer cells. As a result, they often have distinct and less severe side effects compared to chemotherapy. Before administering these medications, cancer cells are examined to determine if they possess the genes, proteins, or other factors that the targeted therapies are designed to address.

Ten years prior, the Food and Drug Administration (FDA) has authorized three various types of targeted therapy for the treatment of EC. These treatments make use of drugs that are designed to target and eliminate cancer cells. Epidermal Growth Factor Receptor (EGFR, Her1, ErbB1), Mesenchymal–Epithelial Transition (MET) factor, Human Epidermal Growth Factor Receptor 2 (HER2, Neu, ErbB2), Vascular Endothelial Growth Factor (VEGF), and Programmed Death Ligand 1 (PD-L1) are examples of targeted therapy drugs. The US Food and Drug Administration (FDA) has approved several targeted therapies for adenocarcinomas [91].

One example is trastuzumab, which was approved in 2010 as the first targeted therapy for gastroesophageal and gastric cancer based on the results of the trastuzumab for gastric cancer (ToGA) study [92,93].

### 4.3.1. Human Epidermal Growth Factor Receptor 2

Human Epidermal Growth Factor Receptor 2 (HER2, Neu, ErbB2) has emerged as a significant target in the treatment of EC [94]. HER2 belongs to the family of EGFR and is a tyrosine kinase which is localized to the plasma membrane. It plays a role in intracellular–extracellular signaling and regulates cell growth, proliferation, differentiation, and cancer development [95]. There is an abnormally high level of the HER2 protein on the surface of cancer cells in some cases of EC, which contributes to cancer cell growth. This protein overexpression is typically caused by an amplification of the HER2 gene. ECs with elevated HER2 levels are referred to as HER2-positive.

Targeting HER2 has become a crucial approach in molecular-based treatment for EC [96]. Lapatinib and trastuzumab are the main therapeutic drugs used for targeting HER2. HER2 overexpression has been observed in 83 percent of ECs, with adenocarcinoma demonstrating higher rates of HER2 positivity compared to squamous cell carcinoma (56 percent) [97]. Various approaches are being explored to target EC effectively.

### Trastuzumab

Trastuzumab is an antibody–drug conjugate that consists of a humanized anti-HER2 antibody, a unique enzyme-cleavable linker, and a topoisomerase I inhibitor. It is a monoclonal antibody, which is a synthetic version of a protein produced by the immune system, and it was the first FDA-approved targeted therapy for HER2-positive gastroe-

sophageal cancer (GE) [98]. Initially approved for the treatment of HER2-positive breast cancer, trastuzumab has shown efficacy in reducing the progression of esophageal tumors with HER2 expression [87]. It can be used to treat certain HER2-positive tumors at the gastroesophageal junction. Trastuzumab is a protein engineered to mimic naturally occurring immune system proteins. While it was originally marketed under the brand name Herceptin, several biosimilars (similar versions) such as Ogivri, Herzuma, Ontruzant, Trazimera, and Kanjinti are now available.

In the treatment of EC, eligible patients typically receive trastuzumab intravenously in combination with chemotherapy at three-week intervals [99]. After chemotherapy, trastuzumab may be administered as a standalone treatment if the cancer responds positively. The medication is continued until there is evidence of cancer recurrence, a phase known as maintenance therapy. Trastuzumab is not used for patients who test negative for HER2 because its mechanism of action specifically targets cancers driven by high levels of the HER2 protein. Research indicates it is not effective against HER2-negative tumors [100]. Trastuzumab may cause mild flu-like symptoms such as fever, vomiting, infusion reactions, diarrhea, fatigue, rash, neutropenia, anemia, headache, and cough. The most severe side effects include cardiomyopathy, pulmonary toxicity, infusion reactions, and febrile neutropenia. It carries a risk of causing heart damage, hence the need for frequent monitoring and testing before initiating treatment to detect potential reactions.

### 4.3.2. Epidermal Growth Factor Receptor

The ErbB family, which also includes the tyrosine kinases ErbB2/HER2/Neu, ErbB3/ HER3, and ErbB4/HER4, includes the Epidermal Growth Factor Receptor (EGFR), also referred to as Her1 or ErbB1. When the EGFR is activated, it causes the receptor to dimerize and become autophosphorylated. This sets off downstream signaling pathways like RAS-RAF-MEK-ERK-MAPK and PI3K-AKT-mTOR, which are crucial for cell proliferation, differentiation, and survival [101]. Cancer development has been linked to abnormal EGFR activation, including overexpression and ligand-dependent receptor heterodimerization [102]. EGFR overexpression has been observed in EC, and clinical trials are currently underway to investigate anti-EGFR agents as potential targeted therapies for this receptor. Over 20 years ago, EGFR targeting was proposed for cancer therapy, and monoclonal antibodies and small-molecule EGFR tyrosine kinase inhibitors (TKIs) are the most promising and extensively studied EGFR-targeting agents [9]. A number of drugs that inhibit EGFR signaling have been developed, providing effective treatment options for EC. Biological agents that target EGFR are being developed as new anticancer therapies. The extracellular region of the EGFR is recognized by monoclonal antibodies like cetuximab and nimotuzumab, which prevent EGF from binding to the receptor and prevent the growth of EC. Small-molecule inhibitors like erlotinib and gefitinib act on the intracellular portion of the receptor to disrupt intracellular tyrosine kinase activity and halt the growth and spread of cancer. Drug resistance resulting from mutations in EGFR-related genes must be taken into account when employing EGFR-targeted therapy for EC. Ongoing research is focused on the development of EGFR inhibitors for gastric cancer and EC, suggesting that targeting EGFR may be a more effective treatment approach. Ongoing research is focusing on the development of EGFR inhibitors for gastric and EC, implying that targeting EGFR may be a more effective treatment strategy.

### Cetuximab

Cetuximab (IMC-C225) is an FDA-approved chimeric monoclonal antibody that targets EGFR in EC treatment. It is derived from a mouse myeloma cell line and belongs to the immunoglobulin G1 (IgG1) class of antibodies. Cetuximab specifically inhibits EGFR activation, thereby slowing down the progression of cancer. It prevents the binding of EGF and TGF-alpha to EGFR and blocks ligand-induced activation of the tyrosine kinase receptor. Additionally, cetuximab promotes the internalization of EGFR, effectively isolating the receptor from ligand interactions. Several studies have demonstrated the efficacy of

cetuximab in treating EC, particularly when used in combination with chemotherapy [103]. T. Ruhstaller and colleagues reported that adding cetuximab as an adjunct therapy to treatments such as radiotherapy, chemotherapy, and surgery significantly reduces the development of regional esophageal squamous cell carcinoma (ESCC) [104].

A recent meta-analysis of randomized controlled trials shows that cetuximab therapy significantly improved response rate and disease control rate for patients with metastatic EC rather than patients with localized EC [103]. Beside this, chemoradiotherapy plus cetuximab improved the trend towards PFS and MFS [105].

In addition to increasing survival, cetuximab also lessens the chance of metastasis and tumor recurrence. A unique method of treating EC uses cetuximab in conjunction with chemotherapy medicines. It is important to note that cetuximab may be ineffective in cancer patients with low EGFR expression. Among all the currently available anti-EGFR monoclonal antibodies, the chimeric IgG1 cetuximab, also known as 'Erbitux' or C225, is the most advanced. Clinical studies have shown that cetuximab improves overall survival when used in combination with radiation therapy for locally advanced disease and chemotherapy for recurrent/metastatic disease.

Nimotuzumab

Nimotuzumab is a fully recombinant, humanized monoclonal antibody that targets EGFR in the treatment of esophageal squamous cell carcinoma (ESCC). It has a lower adverse effect profile compared to cetuximab and panitumumab, with no skin, renal, or gastrointestinal toxicity. EGFR is a promising therapeutic target for ESCC treatment [106]. In comparison to cetuximab, nimotuzumab exhibits lower toxicity and a relatively low incidence of rash. In patients with ESCC, nimotuzumab has shown greater efficacy than conventional chemotherapy. When combined with chemotherapy, nimotuzumab has demonstrated a significant anticancer effect with manageable side effects in advanced EC [107]. Nimotuzumab with chemotherapy has also demonstrated promising therapeutic results in individuals with locally progressed or metastatic ESCC, with little toxicity seen and good tolerance. Therefore, this combined therapy could potentially serve as a novel treatment option for patients with metastatic esophageal squamous cell carcinoma (ESCC) [108].

### 4.3.3. Vascular Endothelial Growth Factor

Tumor angiogenesis includes complex interactions between regulators and effectors. Vascular endothelial growth factors (VEGFs) play an important role in vascular endothelial cell proliferation and angiogenesis among these regulators. VEGFs comprise PlGF, VEGF-A, VEGF-B, VEGF-C, and VEGF-D [109]. A signaling molecule called VEGF is involved in the complex process of tumor angiogenesis. When VEGFs bind to the VEGFRs (VEGFR1, VEGFR2, and VEGFR3), they have an autocrine or paracrine effect. VEGFs are released by tumors or stromal cells. Phosphatidylinositol 3-kinase/protein kinase B (PI3K/AKT) and extracellular regulated protein kinase 1/2 (ERK1/2) are two signaling pathways that can be activated by the interaction of VEGF and its receptors. These pathways can be activated to promote more cell migration, proliferation, and survival. EC progression and diagnosis are linked to VEGF expression [110], thus highlighting the VEGF signaling pathway as a promising therapeutic target for this disease.

Ramucirumab

Ramucirumab is a targeted drug that shows promise in the treatment of certain types of EC. It is a novel anti-angiogenic monoclonal antibody of the human immunoglobulin G (IgG) 1 class, which can block VEGFR2 and hinder its interaction with the ligands. By doing so, it slows down angiogenesis and induces apoptosis of tumor cells [111]. Ramucirumab may be considered as a treatment option when first-line therapies have been ineffective. Its objective is to disrupt angiogenesis, the process of forming new blood vessels. Anti-angiogenesis therapies aim to "starve" the tumor by depriving it of the

nutrients supplied by blood vessels. A protein called VEGF causes the body to produce new blood vessels. Initially, receptors (proteins) are present on the surface of cancer cells—where VEGF binds. Ramucirumab works by preventing VEGF from binding to cancer cells, potentially halting the cancer's progression and slowing its growth [112]. Ramucirumab is used in the diagnosis and management of cancers originating at the gastroesophageal (GE) junction. It is commonly employed as a subsequent treatment option when other drugs have not been successful. It can be administered alone or in combination with the chemotherapy drug paclitaxel. The medication is infused intravenously every two weeks. For advanced ECs that have not responded to previous treatments, ramucirumab could be an alternative option. The most frequent side effects of ramucirumab include fatigue, increased blood pressure, and swelling in the arms or legs. In rare cases, the drug may lead to serious complications such as excessive bleeding, blood clots, and perforation of the digestive system.

*4.4. Immunotherapy for Esophageal Cancer*

Immunotherapy is a groundbreaking class of drugs that stimulates the immune system to fight against cancer. It is a type of cancer treatment that enhances the body's natural defense against cancer cells. Various types of immunotherapy employ different mechanisms of action. Some treatments assist the immune system in halting or slowing down the growth of cancer cells, while others aid in the immune system's destruction of cancer cells or prevention of cancer spread to other parts of the body. Immunotherapy has been predicted to revolutionize modern cancer treatment.

Immunotherapy has significantly improved patients' survival and quality of life compared to previous standards of care, which included radiotherapy, surgery, and chemotherapy [113]. It has firmly established itself as a new cornerstone of cancer care across various cancer types, from metastatic to adjuvant and neoadjuvant settings. Immunotherapy, either as a standalone treatment or in combination with chemotherapy, has proven to be effective. Recent studies have demonstrated remarkable response rates in advanced and resectable EC patients using checkpoint inhibitors such as nivolumab and pembrolizumab [114]. While the current results from large clinical trials show great efficacy with manageable toxicity, favorable survival rates, and long-term quality of life, some concerns still remain. Overall, immunotherapy represents a promising avenue for cancer treatment, offering new hope for patients. Ongoing research aims to refine and optimize immunotherapy approaches, expand its applicability, and address any remaining challenges to further enhance patient outcomes.

4.4.1. Immune Checkpoint Inhibitors

The immune system is a highly specialized and complex biological network consisting of specific cells, organs, and proteins. It can be divided into two types: innate (non-specific) immunity and adaptive (specific) immunity. Innate immune components such as natural killer (NK) cells, macrophages, and dendritic cells play a crucial role in identification and containment of cancer cells. However, to recognize and eliminate tumor cells, T cells from the adaptive immune system are recruited [115]. The immune response within the body is regulated by co-stimulatory and inhibitory signals under normal physiological conditions. When the immune system is triggered by signals from pathogenic stimuli, it can precisely detect and eliminate specific antigens. Normal tissues protect the immune system from self-destruction by expressing immunological checkpoints known as self-tolerance. Immune cells play a significant role in suppressing tumor cells, and it has been observed that individuals with weakened immune systems are more susceptible to developing cancer. Cancer cells employ various strategies to evade detection by the immune system and to manipulate immune cells. These strategies are collectively termed immune evasion.

Immune checkpoints are crucial for preventing autoimmune reactions by blocking T-cell receptors (TCRs) from recognizing the antigens [116]. Immune checkpoint proteins prevent tumor-specific T cells from performing their function, allowing cancer cells to

avoid immune surveillance. Cancer cells express immune checkpoint proteins like Programmed Cell Death 1 Ligand 1 (PD-L1, also known as CD279) and receptors like cytotoxic T lymphocyte-associated antigen-4 (CTLA-4, also known as CD152), which have been linked to the suppression and downregulation of T-cell activity. Cancer cells additionally secrete exosomes which contain these immune checkpoint regulators [117]. These checkpoint regulators deliver inhibitory signals that suppress the immune response when they bind to proteins expressed on immune cells (T cells, B cells, and myeloid cells). Additionally, recent studies have shown that cancer cells release extracellular vesicles (EVs) that express the protein PD-L1 [118]. The immune system is a complex network of communicating cells and biochemical signals that coordinates the identification and eradication of foreign antigens while guarding against autoimmune reactions [119]. Finely controlled connections between immune cells and a dynamic interplay of stimulatory and inhibitory signals keep this delicate balance in check [120]. In order to avoid immune surveillance, cancer cells frequently modify signaling pathways to upset this equilibrium. To combat tumor immune evasion, one can use stimulatory receptor antagonists or inhibitory signal antagonists, effectively harnessing the immune system as a weapon against cancer [121]. Both approaches enhance the specific anti-cancer action of the immune system. Furthermore, current research suggests that immunotherapy may benefit only a subset of EC patients [122]. The detailed mechanism is described in Figure 1.

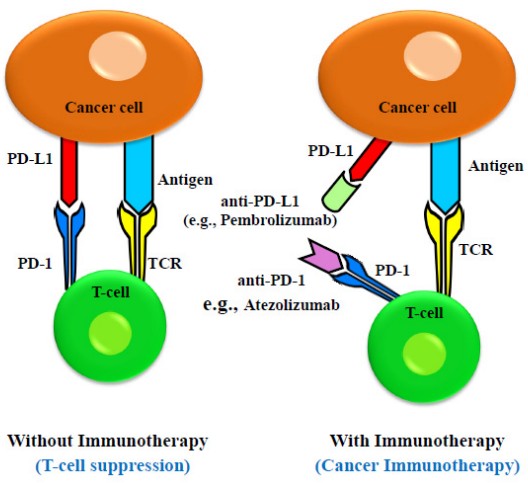

**Figure 1.** Schematic drawing of molecular mechanism of anti-PD-1 and anti-PD-L1 inhibitor-mediated cancer immunotherapy. Immune effect of T cell is suppressed when there is an interaction between PD-1 on T-cell surface and PD-L1 on cancer cells. The therapeutic antibodies (anti-PD-1 and anti-PD-L1) for EC immunotherapy bind to PD-1 and PD-L1 and inhibit their interaction.

Nivolumab

The FDA approved nivolumab in 2020 for patients with unresectable advanced, recurring, or metastatic esophageal squamous cell carcinoma. Nivolumab is a completely human IgG4 monoclonal antibody that targets PD-1. It blocks the binding of PD-L1 and PD-L2 to PD-1, and prevents immunosuppression [123]. As a member of the CD28 superfamily, PD-1 is a crucial immunosuppressive protein which restricts T-cell activation when it interacts with PD-L1 and PD-L2, two of its ligands. Numerous studies have shown significant PD-L1 and PD-L2 expression in EC, with PD-L1 expression linked to tumor invasion depth and a bad prognosis and PD-L2 expression linked to reduced CD8+ T-cell infiltration. The PD-1 receptor is highly expressed on T cells, B cells, and NK cells. The PD-L1 receptor's ligand, also known as CD274, is called the PD-1 receptor. When exposed to inflammatory cytokines, this molecule, which is expressed in peripheral organs, suppresses T-cell function. Additionally, PD-L1 on activated mature NK cells is upregulated by the inflammatory cytokine interleukin (IL)-18, which contributes to immunosuppression [124].

In melanoma, lung, breast, pancreatic, gastric, colon, ovarian, and ECs, PD-L1 is typically overexpressed on cancer cells. This makes it possible for tumor cells to interact with T-cell PD-1 receptors, inhibiting T-cell activation, proliferation, and ultimately T-cell death [125]. It is widely known that the expression of PD-L1 is a key mechanism by which many types of cancer evade the immune system. In light of this, it is not unexpected that PD-L1 and PD-1 inhibitors have become popular and highly successful cancer immunotherapies. This data emphasizes how important PD-1 inhibition is for the treatment of EC [126].

Pembrolizumab

The US Food and Drug Administration (FDA) authorized pembrolizumab, a monoclonal antibody that targets PD-1, as the first immune checkpoint inhibitor in 2014 for the treatment of esophageal and esophagogastric junction cancer [127]. Patients who have already received conventional chemotherapy and have locally progressed or metastatic squamous cell EC can benefit from pembrolizumab, commonly known by the brand name Keytruda, which is a trusted and safe immunotherapy medication [128]. T cells, which are immune system cells responsible for eradicating foreign invaders and aberrant cells in the body, such as cancer cells, attach to PD-1 proteins on PD-L1 proteins. Because this attachment prevents T cells from performing their immunological job, cancer cells are able to avoid being recognized by the immune system. When pembrolizumab binds to PD-1, it stops PD-L1 from interacting with it [129].

Pembrolizumab was approved by the FDA as a second-line treatment for patients with squamous EC, which develops in the cells lining the esophagus and spreads to nearby lymph nodes, local areas, and beyond. The results of a clinical trial were published on 6 October in the Journal of Clinical Oncology, supporting the FDA's decision. The drug's target protein, Programmed Death Ligand-1 (PD-L1), must also be present in significant concentrations in the cancer cells of the patients [123]. Pembrolizumab, a PD-1 inhibitor, is well tolerated in EC patients who are PD-L1 positive, according to preliminary results from ongoing studies.

As a result of immunotherapy, the survivability of the cancer patients becomes considerably higher. Toxicities associated with immunotherapy for EC are comparatively uncommon. Immuno-oncology is still in its early phases and confronts numerous difficulties despite the quick progress in the field. Finding biomarkers that can forecast the success of immunotherapy is therefore essential [130]. This would make it possible to pick the individuals who would benefit from the therapy while protecting those who would not from its side effects and failures, like cutaneous, gastrointestinal, endocrine, and liver toxicity. The effectiveness of various immunotherapies in combination with other EC therapy options is now being examined in numerous clinical trials with the goal of enhancing therapeutic alternatives for patients with this condition.

4.4.2. Toxicities Associated with Immunotherapy

The frequency and severity of immune checkpoint inhibitor (ICI)-related toxicity are still being investigated, and much of the available data is based on trials involving ipilimumab, pembrolizumab, and nivolumab. The adverse effects associated with immunotherapy have been a significant hurdle for its widespread use, and the development of immune-related adverse events (irAEs) is linked to the reduction of immunosuppression [131]. The most commonly observed immune-related adverse effects include endocrine, skin, liver damage, and gastrointestinal issues. However, comprehensive data on irAEs for newer drugs is still being collected and evaluated. Due to the nature of irAEs and underreporting, the reported rates are likely to underestimate the true occurrence of these events [132].

The estimated prevalence of any-grade irAEs associated with single-agent ICI therapy varies widely across different drugs and trials, ranging from approximately 15% to 90%. Since these irAEs can occur at any stage, close monitoring, timely follow-up, and careful

management are necessary [133]. Therefore, physicians must be skilled in effectively diagnosing and managing these side effects.

## 5. Common Treatment Approaches

### 5.1. Cancer Treatment by Stage

After the discovery of EC, doctors will try to determine whether it has spread and, if so, how far it has spread. This process is referred to as staging. The stage of cancer indicates how much of it is present in the body and helps determine the severity of the cancer and the most effective course of treatment [134].

The evaluation of illness extent at the time of initial diagnosis is critical for establishing effective management, subsequent therapy outcomes, and prognosis. To closely correlate stage and disease prognosis, a revised tumor–node–metastases (TNM) primary tumor (T), lymph node involvement (N), and extent of metastatic disease (M) classification was used in 1988; however, recent literature suggests that this TNM system did not fully distinguish staging according to survival [135]. Depth of wall penetration and lymph node metastases were found to be stronger prognostic indicators, and the American Joint Committee on Cancer (AJCC) changed the staging system to include these prognostic characteristics in 2002 [136]. The initial stage of EC is known as stage 0 (high-grade dysplasia). It then progresses from stage I (1) to stage IV (4). The innermost layer of the lining of the esophagus, the epithelium, is where the majority of the ECs begins and slowly progress to innermost layers.

The American Joint Committee on Cancer (AJCC) TNM method, which is based on three important pieces of evidence, is the most often used staging approach for EC [137]. Grade—The grade is another aspect that can influence the therapy procedure [138]. When viewed through a microscope, this value ranges from 1 to 3. For details, see Tables 2 and 3.

**Table 2.** Esophageal cancer (adenocarcinoma) stage grouping illustrated in 7th edition of American Joint Committee on Cancer (AJCC) Cancer Staging Manual.

| Stage | T | N | M | Grade |
|:-----:|:---:|:---:|:---:|:-----:|
| 0 | Is | 0 | 0 | 1 |
| IA | 1 | 0 | 0 | 1–2 |
| IB | 1 | 0 | 0 | 3 |
|  | 2 | 0 | 0 | 1–2 |
| IIA | 2 | 0 | 0 | 3 |
| IIB | 3 | 0 | 0 | Any |
|  | 1–2 | 1 | 0 | Any |
| IIIA | 1–2 | 2 | 0 | Any |
|  | 3 | 1 | 0 | Any |
|  | 4a | 0 | 0 | Any |
| IIIB | 3 | 2 | 0 | Any |
| IIIC | 4a | 1–2 | 0 | Any |
|  | 4b | Any | 0 | Any |
|  | Any | 3 | 0 | Any |
| IV | Any | Any | 1 | Any |

**Table 3.** Esophageal cancer (squamous cell carcinoma) stage grouping illustrated in 7th edition of American Joint Committee on Cancer (AJCC) Cancer Staging Manual.

| Stage | T | N | M | Grade |
|---|---|---|---|---|
| 0 | 1 | 0 | 0 | 1 |
| IA | 1 | 0 | 0 | 1 |
| IB | 1 | 0 | 0 | 2–3 |
| | 2–3 | 0 | 0 | 1 |
| IIA | 2–3 | 0 | 0 | 1 |
| | 2–3 | 0 | 0 | 2–3 |
| IIB | 2–3 | 0 | 0 | 2–3 |
| | 1–2 | 1 | 0 | Any |
| IIIA | 1–2 | 2 | 0 | Any |
| | 3 | 1 | 0 | Any |
| | 4a | 0 | 0 | Any |
| IIIB | 3 | 2 | 0 | Any |
| IIIC | 4a | 1–2 | 0 | Any |
| | 4b | Any | 0 | Any |
| | Any | 3 | 0 | Any |
| IV | Any | Any | 1 | Any |

GX: The grade cannot be assessed. (The grade is unknown).

Grade 1 (G1: well differentiated; low grade): Cancer cells closely resemble normal esophageal cells.

Grades 2 (G2: moderately differentiated; intermediate) comes in between Grades 1 and 3.

Grade 3 (G3: poorly differentiated, undifferentiated; high grade): Cancer cells appear highly abnormal.

*5.2. Treating Esophageal Cancer by Stage*

Stage 0: High-Grade Dysplasia Limited to the Epithelium.

The cancerous cells are confined to the epithelium, the top layer of cells lining the inside of the esophagus, at this stage. There is no invasion into deeper layers, and no spread to lymph nodes or distant organs. Endoscopic procedures such as radiofrequency ablation (RFA), endoscopic mucosal resection (EMR), and photodynamic therapy (PDT) are commonly used to treat this stage. These techniques aim to remove or destroy the abnormal cells. Following endoscopic treatment, regular and thorough follow-up with frequent upper endoscopy is crucial to detect any recurrence of pre-cancerous or cancerous cells in the esophagus [139].

Alternatively, an esophagectomy may be considered as another treatment option. This surgical procedure involves the removal of the abnormal part of the esophagus. While it is a major surgical intervention, one advantage is that it eliminates the need for lifelong endoscopy monitoring.

Proper management and close monitoring are essential in ensuring effective treatment outcomes and reducing the risk of disease progression or recurrence.

Treatment Options for Stage I Esophageal Cancer

Stage I EC refers to very early cancers that are limited to a small area of the mucosa and have not spread to the submucosa. The treatment options for stage I EC depend on various factors, including the extent of the cancer and the overall health of the patient.

*5.3. Endoscopic Treatment*

Some T1a tumors can be treated with endoscopic mucosal resection (EMR), which removes the cancerous area of the esophageal lining. After that, another kind of endoscopic procedure, like ablation, might be performed to eliminate any remaining abnormal areas in the esophageal lining [140].

Surgery (esophagectomy): In most cases, patients with T1 cancer who are in good overall health will undergo surgery (esophagectomy) to remove the cancer.

Pre-surgical chemotherapy: If the cancer is located in the esophagus near the stomach, chemotherapy without radiation is used to treat the cancerous portion of their esophagus.

Adjuvant chemoradiation: If there are signs that not all of the cancer has been removed following surgery, chemotherapy and radiation therapy may be recommended [6].

Neoadjuvant chemoradiation: Neoadjuvant chemoradiation is frequently used before surgery in individuals with T2 tumors that have spread to the muscularis propria [141].

Surgery alone: For smaller tumors (less than 2 cm), surgery alone may be an option.

Immunotherapy: In cases where lab tests following chemoradiation and surgery reveal that some cancer remains, treatment with an immunotherapy medicine, such as nivolumab, may be considered.

Primary chemoradiation: when the cancer is located in the upper region of the esophagus (in the neck), chemoradiation may be used first instead of surgery.

Close monitoring with endoscopy is critical for detecting any signs of cancer recurrence. For individuals who are unable to have or ineligible for surgery due to other major health issues, alternative treatments such as EMR and endoscopic ablation, chemotherapy, radiation therapy, or a combination of these approaches may be considered [53]. The treatment plan should be tailored to the individual patient's circumstances and discussed with a medical professional.

## 6. Treating Stages II and III Cancer of the Esophagus

Cancers in stage II have spread to the esophageal main muscle layer or the connective tissue on the exterior of the esophagus. At this stage, some tumors have also spread to one or two nearby lymph nodes. Cancer has spread through the esophageal wall to the outer layer in stage III, and malignancies have spread to neighboring organs or tissues. Chemoradiation (chemotherapy combined with radiation therapy), followed by surgery, is the most typical treatment for these tumors for those in sufficient health. If laboratory testing reveals that some cancer remains after surgery, treatment with an immunotherapy medicine like nivolumab may be considered.

Patients with gastroesophageal junction adenocarcinoma are sometimes treated with chemotherapy (without radiation) followed by surgery [142]. In certain instances, particularly for upper ECs, chemoradiation may be recommended as the primary treatment rather than surgery [143]. Individuals who do not have surgery require continuous endoscopy monitoring to detect any signs of residual malignancy. Cancer can still exist beneath the inner lining of the esophagus even when it is not visible, so close monitoring is required.

Chemoradiation is frequently used to treat patients who are unable to have surgery due to other major health issues or because the cancer is too large to be removed. Chemotherapy, immunotherapy, or a combination of the two may be used in the absence of chemoradiation. The first line of treatment for people with HER2-positive gastroesophageal junction cancer may consist of immunotherapy with pembrolizumab, followed by chemotherapy and the targeted drug trastuzumab [144].

## 7. Treating Stage IV of Esophagus Cancer

Stage IV EC has spread to other organs or distant lymph nodes. Surgery is typically not a viable option for trying to cure these malignancies because they are typically very difficult to completely eradicate. The primary goal of treatment is to control the cancer for as long as possible and alleviate any symptoms it may cause. To enhance patient well-being

and lengthen survival, chemotherapy may be given, possibly in combination with targeted medication therapy or immunotherapy.

Radiation therapy or other therapies may be performed to alleviate discomfort or difficulties in swallowing. Targeted treatments like larotrectinib or entrectinib may be considered if the cancer cells have particular gene alterations [145].These treatments aim to target the specific genetic abnormalities in the cancer cells.

## 8. Post-Operative Complications and Rehabilitation Care for Esophageal Cancer

EC is not only difficult to treat but is also often accompanied by a range of complications. The presence of multiple complications makes EC a highly lethal disease that poses significant challenges for treatment, causing ongoing harm to patients. Currently, surgery remains the primary therapy for individuals with EC. However, surgical procedures and postoperative therapies can lead to complications associated with EC [46].

In the past, EC had a dismal prognosis, but survival rates have been improving. Worldwide, the 5-year survival rates for localized cancers have increased to around 40% [146]. Surgery, specifically esophagectomy, is often combined with a multimodality approach that includes neoadjuvant chemotherapy or chemoradiotherapy to treat EC. However, esophagectomy carries a significant risk of morbidity and a mortality rate of up to 5% [147].

Oncologic esophagectomy for cancer is a complex and demanding procedure involving both the abdomen and thoracic region, which results in chronic surgical trauma. This can have adverse effects on the physiological condition and physical integrity of patients. Long-term consequences of EC include fatigue, reflux, dysphagia, discomfort, and diarrhea [148], which can lead to nutritional and functional deficits and a lower quality of life [149]. Malnutrition is a serious long-term problem for individuals who have undergone esophagectomy [150]. Other issues such as dysphagia and reflux can also impede a patient's ability to consume an adequate amount of food [151]. Weight loss after surgery is characterized by the loss of fat mass and depletion of skeletal muscle mass, a condition known as sarcopenia [152,153]. Sarcopenia has been identified as a potential biomarker for poor prognosis in patients with EC [154].

Patients undergoing EC surgery are at a considerable risk of experiencing postoperative decline in lung function, leading to pulmonary complications. One-lung ventilation during thoracotomy/thoracoscopy contributes to this risk. It is a major factor in prolonged stays in the intensive care unit, delayed postoperative recovery, and diminished quality of life. Potential postoperative surgical complications include pulmonary and cardiovascular events, thromboembolism, infectious events, as well as the risk of anastomotic leak, thoracogastric fistula, conduit necrosis, chyle leak, chylothorax, reflux esophagitis, anastomotic stenosis, severe diarrhea, and vocal cord injury/palsy, all of which can be fatal [155].

### 8.1. Pulmonary Complication

The integrity of the thoracic wall and intercostal muscles, notably the diaphragm, is weakened even though the lung tissue is not removed during the EC surgery. Infections of the respiratory tract are frequently brought on by this damage to the lung's ventilation pump. Patients who feel pain from wounds in the neck, chest, or upper abdomen after surgery, or if the stomach is dragged into the chest, compressing the lungs, may experience varied degrees of dyspnea and shortness of breath. Alveolar collapse, pulmonary edema, compromised pulmonary defense mechanisms, and insufficient ventilation are examples of the pathophysiological pathways that contribute to lung infections [156]. Persistent chronic bronchitis and chronic cardiac insufficiency are potential perioperative causes of surgical lung infections [157]. Recent studies have shown that 28% of 2704 individuals experienced serious respiratory problems, 1.5% developed pneumonia, and 7% experienced respiratory failure [158].

*8.2. Reflux Esophagitis*

A frequent postoperative consequence of EC is reflux esophagitis. Body flexion after consuming acidic foods or beverages, as well as food reflux from the stomach and esophagus to the mouth or pharynx while patients are sleeping, are characteristics of the condition. Additionally, this has symptoms like post-sternal burning, soreness, difficulty swallowing, and more. This symptom may be associated with gastrin concentration and vagotomy [159].

*8.3. Functional Gastric Emptying Disorder (FDGE)*

When removing the esophagus in patients with EC, it may also be necessary to remove a portion of the gastric wall or even a section of the stomach itself. Since the esophagus and stomach are interconnected, their functions can be influenced by each other. Resection of EC can lead to gastric motility abnormalities, which can cause dysfunction in the reservoir function of the stomach. Reports indicate that more than half of individuals who undergo esophagectomy experience symptoms of functional dyspepsia with gastric emptying disorder [160].

There is a recognized need for research on rehabilitation programs for less-studied diseases like EC in order to address the diverse rehabilitative needs that survivors may face. Currently, research on EC rehabilitation has focused on prehabilitation, neoadjuvant treatment, and early postoperative inpatient rehabilitation [161].

## 9. Conclusions

In conclusion, EC stands out as a significant public health concern, necessitating a multidisciplinary approach for its treatment. This review summarizes various therapies, including endoscopic treatment, surgery, chemoradiotherapy, immunotherapy, and targeted therapy. Among these, targeted therapy exhibits promise in addressing metastatic EC. The development of targeted drugs with both high efficacy and minimal side effects holds pivotal importance for the effective treatment of EC. Further research in this field remains essential. Overall, a comprehensive approach that integrates multiple therapeutic modalities is imperative to enhance outcomes for patients with EC.

**Author Contributions:** R.A., A.M. and H.K.V. were involved in the data collection and validation; provided the first draft of the manuscript; R.A., A.M., H.K.V. and L.V.K.S.B. prepared the figures and tables, wrote and finalized the manuscript; H.K.V. and L.V.K.S.B. designed the outline and coordinated the writing of the paper. All authors have read and agreed to the published version of the manuscript.

**Funding:** This research received no external funding.

**Institutional Review Board Statement:** Not applicable.

**Informed Consent Statement:** Not applicable.

**Data Availability Statement:** Not applicable.

**Conflicts of Interest:** The authors declare no conflict of interest.

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
