# Peer review of "Unveiling Therapeutic Targets for Esophageal Cancer: A Comprehensive Review"

_curroncol, doi:10.3390/curroncol30110691_

Round 1

Reviewer 1 Report

Comments and Suggestions for Authors

The manuscript “Unveiling Therapeutic Targets for Esophageal Cancer: A Comprehensive Review” is a review article regarding the actual state-of-art of traditional and novel therapies, including endoscopic treatment, surgery, chemoradiotherapy, immunotherapy, and targeted therapy for Esophageal Cancer management.

The manuscript may be of interest for the readers. However, there are major concerns that must be addressed in order to consider the manuscript suitable for publication.

Major

1.       A language revision is mandatory. The manuscript is full of mistakes and typos (e.g. table “definations”;

2.       The introduction section should be improved providing a larger background about Esophageal Cancer epidemiology.

3.       The manuscript contains weird things, such as in line 743; there is written Table 1 without reason.

4.       I do not understand why some parts are highlighted in yellow or in red font.

5.       The quality of figure 1 is really poor. Please improve it. There is also the red sign for the spelling error under two words which must be deleted.

6.       The manuscript lacks of important findings available in literature. For instance, authors should report that a strategy is based on drug repositioning approach (PMID: 36246638).

7.       Analogously, another promising therapeutic target in Esophageal Cancer is the enzyme nicotinamide N-methyltransferase (NNMT) which is overexpressed in this neoplasm and promotes the aggressiveness (PMID: 36139012). A number of NNMT inhibitors are already available and could be tested for Esophageal Cancer management (PMID: 34572571; PMID: 34704059; PMID: 34424711).

8.       The subparagraphs in the section “systemic treatment” are not consistent in form and style.

9.       Figure 2 is confused and difficult to be understood by readers. Please rearrange it in a clearer way and improve quality.

Comments on the Quality of English Language

Extensive editing of English language required

Author Response

Reviewer 1

The manuscript “Unveiling Therapeutic Targets for Esophageal Cancer: A Comprehensive Review” is a review article regarding the actual state-of-art of traditional and novel therapies, including endoscopic treatment, surgery, chemoradiotherapy, immunotherapy, and targeted therapy for Esophageal Cancer management.

The manuscript may be of interest for the readers. However, there are major concerns that must be addressed in order to consider the manuscript suitable for publication.

Major

  1. A language revision is mandatory. The manuscript is full of mistakes and typos (e.g. table “definations”

Answer: Thank you for the suggestion we have modified the manuscript with language correction.

  1. The introduction section should be improved providing a larger background about Esophageal Cancer epidemiology.

Answer: Thank you for the suggestion. We have provided enough information regarding the esophageal cancer epidemiology part.

  1. The manuscript contains weird things, such as in line 743; there is written Table 1 without reason.

Answer: Thank you for the suggestion. We have removed this from the manuscript.

  1. I do not understand why some parts are highlighted in yellow or in red font.

Answer: Thank you for the suggestion. We are not able to find any yellow or red highlighted font.

  1. The quality of figure 1 is really poor. Please improve it. There is also the red sign for the spelling error under two words which must be deleted.

Answer: Thank you for the suggestion. We have removed the red sign for the spelling error.

  1. The manuscript lacks of important findings available in literature. For instance, authors should report that a strategy is based on drug repositioning approach (PMID: 36246638).

Answer: As per the suggestion, we have incorporated the drug repositioning approach.

  1. Analogously, another promising therapeutic target in Esophageal Cancer is the enzyme nicotinamide N-methyltransferase (NNMT) which is overexpressed in this neoplasm and promotes the aggressiveness (PMID: 36139012). A number of NNMT inhibitors are already available and could be tested for Esophageal Cancer management (PMID: 34572571; PMID: 34704059; PMID: 34424711).

Answer: We have incorporated all the information as per the recommendation.

  1. The subparagraphs in the section “systemic treatment” are not consistent in form and style.

Answer: Thank you for the suggestion. The subparagraphs in the section “systemic treatment” are now consistent in form.

  1. Figure 2 is confused and difficult to be understood by readers. Please rearrange it in a clearer way and improve quality.

Answer: Thank you for the suggestion. We have removed Figure 2

Reviewer 2 Report

Comments and Suggestions for Authors

In the present review, “Unveiling Therapeutic Targets for Esophageal Cancer: A Comprehensive Review," the authors aim to comprehensively understand the therapeutic approaches employed in managing esophageal cancer. The review on this topic is of great importance to the field; however, some issues and considerations need to be reviewed and answered by the authors.

General issues

1- The presentation of the topics is somewhat confusing; it is unclear what the main items and sub-items are. Furthermore, some issues appear to be out of order.

2- I recommend starting the text from the end by introducing the staging and then structuring the entire text to discuss the treatments accordingly.

3- I would recommend that the authors provide further clarification in all sections, particularly in those discussing therapies and treatments, regarding the distinctions (where they exist) between SCC and adenocarcinoma.

4- There are highlighted sentences or paragraphs throughout the text, sometimes written in red letters and at other times with yellow highlighting. Please make the necessary corrections.

5- I suggest that the authors review the entire text, as there are sentences written in a language that is too informal for a scientific paper, e.g.: lines 150-152 and 728-729.

Specific issues

1- line 36. Please review this sentence, as most references, including the one mentioned by the authors, consider that ESCC affects approximately 90% of the world's population, not 70%.

2- lines 48-49. Please reconsider this information, as most studies do not demonstrate the advantages in using tyrosine kinase inhibitors to treat esophageal cancer. In the editorial cited by the authors, it is mentioned that: "Given the consistent and negative results for EGFR-targeted therapy in esophagogastric cancer, outside of a biomarker driven clinical trial, it is not justifiable to move these agents forward. In this context, the SAKK trial has to be viewed as an outlier. At the end of the day, the absolute number of patients achieving either a local control benefit (6 of 188 adenocarcinoma and 8 of 109 squamous cancer patients) or a survival benefit (8 of 188 adenocarcinoma and 9 of 109 squamous cancer patients) was quite small.”

3- lines 462. In this section, the authors should explicitly clarify that the benefit of this treatment applies to patients with tumors displaying high HER-2 expression. Additionally, within this section, the authors need to include the percentage of patients with esophageal cancer who exhibit this HER2 overexpression.

4- line 477. The Trastuzumab for Gastric Cancer (ToGA) study aimed to evaluate the clinical effectiveness and safety of combining trastuzumab with first-line chemotherapy for advanced gastric or gastroesophageal junction cancers exhibiting HER2 overexpression. Please make this more evident in the text.

5- line 501. The authors should explicitly state that the benefit of this treatment is most pronounced in patients with tumors exhibiting high EGFR expression. Furthermore, within this context, the authors should incorporate the percentage of esophageal cancer patients displaying this super expression of EGFR, who could potentially derive benefits.

6- lines 502-504. This sentence should precede this item, as the authors mentioned HER2 in the previous section.

7. line 526. The authors should discuss in the text that most clinical trials using cetuximab added to multimodal therapy have demonstrated improvements in response and disease control rates in patients with metastatic esophageal tumors. They should also note that most studies did not significantly improve overall and progression-free survival for patients with localized or metastatic esophageal cancer.

8. lines 584-585. The authors should provide a more detailed discussion of this information, as not all types of esophageal cancer may benefit from this treatment.

9. lines 653-654. Are all patients in this group eligible for nivolumab treatment, or only those who express PD-1?

Author Response

Reviewer 2

In the present review, “Unveiling Therapeutic Targets for Esophageal Cancer: A Comprehensive Review," the authors aim to comprehensively understand the therapeutic approaches employed in managing esophageal cancer. The review on this topic is of great importance to the field; however, some issues and considerations need to be reviewed and answered by the authors.

General issues

1- The presentation of the topics is somewhat confusing; it is unclear what the main items and sub-items are. Furthermore, some issues appear to be out of order.

Answer: Thank you for the suggestions. The main items and the sub-items are now clearly marked and are in order.

2- I recommend starting the text from the end by introducing the staging and then structuring the entire text to discuss the treatments accordingly.

Answer: Thank you for the suggestions, but we have already mentioned it in the introduction section.

3- I would recommend that the authors provide further clarification in all sections, particularly in those discussing therapies and treatments, regarding the distinctions (where they exist) between SCC and adenocarcinoma.

Answer: Thank you for the in-deep analysis. We have highlighted all recommendation in treatment option.

4- There are highlighted sentences or paragraphs throughout the text, sometimes written in red letters and at other times with yellow highlighting. Please make the necessary corrections.

Answer: Thank you for the suggestion. There are no such highlighted sentences or paragraphs. We made the necessary corrections.

5- I suggest that the authors review the entire text, as there are sentences written in a language that is too informal for a scientific paper, e.g.: lines 150-152 and 728-729.

Answer: Thank you for the suggestion. We have reviewed the entire manuscript and the language issues were fixed wherever required.

Specific issues

1- line 36. Please review this sentence, as most references, including the one mentioned by the authors, consider that ESCC affects approximately 90% of the world's population, not 70%.

Answer: As per the suggestion, we have corrected it by analysing recent literatures.

2- lines 48-49. Please reconsider this information, as most studies do not demonstrate the advantages in using tyrosine kinase inhibitors to treat esophageal cancer. In the editorial cited by the authors, it is mentioned that: "Given the consistent and negative results for EGFR-targeted therapy in esophagogastric cancer, outside of a biomarker driven clinical trial, it is not justifiable to move these agents forward. In this context, the SAKK trial has to be viewed as an outlier. At the end of the day, the absolute number of patients achieving either a local control benefit (6 of 188 adenocarcinoma and 8 of 109 squamous cancer patients) or a survival benefit (8 of 188 adenocarcinoma and 9 of 109 squamous cancer patients) was quite small.”

Answer: Thank you very much the in-deep analysis, we could not find the sentence in the manuscript could you please highlight in the manuscript that would be great for the revision.

3- lines 462. In this section, the authors should explicitly clarify that the benefit of this treatment applies to patients with tumors displaying high HER-2 expression. Additionally, within this section, the authors need to include the percentage of patients with esophageal cancer who exhibit this HER2 overexpression.

Answer: As per the suggestion, the percentage of patients with EC who exhibit the HER2 overexpression are verified.

4- line 477. The Trastuzumab for Gastric Cancer (ToGA) study aimed to evaluate the clinical effectiveness and safety of combining trastuzumab with first-line chemotherapy for advanced gastric or gastroesophageal junction cancers exhibiting HER2 overexpression. Please make this more evident in the text.

Answer: We appreciate the suggestion. But we found limited information regarding this.

5- line 501. The authors should explicitly state that the benefit of this treatment is most pronounced in patients with tumors exhibiting high EGFR expression. Furthermore, within this context, the authors should incorporate the percentage of esophageal cancer patients displaying this super expression of EGFR, who could potentially derive benefits.

Answer: Thank you for the recommendation, but we were unable to find the exact percentage of esophageal cancer patients displaying this super expression of EGFR.

6- lines 502-504. This sentence should precede this item, as the authors mentioned HER2 in the previous section.

Answer: Thank you for the suggestion. But the above information between the lines 502-504 is necessary at that position.

  1. line 526. The authors should discuss in the text that most clinical trials using cetuximab added to multimodal therapy have demonstrated improvements in response and disease control rates in patients with metastatic esophageal tumors. They should also note that most studies did not significantly improve overall and progression-free survival for patients with localized or metastatic esophageal cancer.

Answer: Thank you for the suggestion. We have included required information in the manuscript.

  1. lines 584-585. The authors should provide a more detailed discussion of this information, as not all types of esophageal cancer may benefit from this treatment.

Answer: Thank you for the suggestion. In the present manuscript we have tried to include all relative therapeutic target for the EC that will give benefit to clinician and reasercher.

  1. lines 653-654. Are all patients in this group eligible for nivolumab treatment, or only those who express PD-1?

Answer: Not all the patients in this group are eligible for nivolumab treatment this is only for those who express PD-1.

Reviewer 3 Report

Comments and Suggestions for Authors

Over the past few years, I have received many reviews for review on the treatment of certain oncological diseases; each review had a significant number of shortcomings, which, to my surprise, this review does not have. This is a rare case when I recommend that the editor accept this review as presented, and I want to thank the authors for their work and excellent systematization of the material. I believe that this review will be useful both to researchers in the field of developing treatment regimens for cancer patients and to practicing oncologists.

Author Response

Reviewer 3

Over the past few years, I have received many reviews for review on the treatment of certain oncological diseases; each review had a significant number of shortcomings, which, to my surprise, this review does not have. This is a rare case when I recommend that the editor accept this review as presented, and I want to thank the authors for their work and excellent systematization of the material. I believe that this review will be useful both to researchers in the field of developing treatment regimens for cancer patients and to practicing oncologists.

Answer: Thank you very much for the appreciations

Round 2

Reviewer 1 Report

Comments and Suggestions for Authors

The authors improved the quality of the manuscript therefore it can be accepted for publication.

Comments on the Quality of English Language

Moderate editing of English language required

Reviewer 2 Report

Comments and Suggestions for Authors

Dear authors,

after reviewing the revised text and incorporating additional considerations, I believe the work is now ready for publication.

Kind regards